# Some Magnetic Properties and Magnetocaloric Effects in the High-Temperature Antiferromagnet YbCoC$_2$

**Denis Alexandrovich Salamatin** [1,2,*], **Vladimir Nikolaevich Krasnorussky** [1],
**Mariya Viktorovna Magnitskaya** [1], **Alexei Valeryevich Semeno** [1,3], **Alexander Vladimirovich Bokov** [1],
**Atanas Velichkov** [2,4], **Zbigniew Surowiec** [2,5] **and Anatoly Vasilyevich Tsvyashchenko** [1]

[1]   Vereshchagin Institute for High Pressure Physics, RAS, 108840 Moscow, Russia;
      magnma@yandex.ru (M.V.M.); tsvyash@hppi.troitsk.ru (A.V.T.)
[2]   Joint Institute for Nuclear Research, 141980 Dubna, Russia; zbigniew.surowiec@mail.umcs.pl (Z.S.)
[3]   Prokhorov General Physics Institute, RAS, 119991 Moscow, Russia
[4]   Institute for Nuclear Research and Nuclear Energy, 1784 Sofia, Bulgaria
[5]   Institute of Physics, M. Curie-Sklodowska University, 20-031 Lublin, Poland
*   Correspondence: dasalam@gmail.com

**Abstract:** The YbCoC$_2$ compound, which crystallizes in a base-centered orthorhombic unit cell in the *Amm*2 space group CeNiC$_2$ structure, is unique among Yb-based compounds due to the highest magnetic ordering temperature of $T_N = 27$ K. Magnetization measurements have made it possible to plot the *H-T* magnetic phase diagram and determine the magnetocaloric effect of this recently discovered high-temperature heavy-fermion compound, YbCoC$_2$. YbCoC$_2$ undergoes spin transformation to the spin-polarized state through a metamagnetic transition in an external magnetic field. The transition is found to be of the first order. The dependencies of magnetic entropy change $\Delta S_m(T)$—have segments with positive and negative magnetocaloric effects for $\Delta H \leq 6$ T. For $\Delta H = 9$ T, the magnetocaloric effect becomes positive, with a maximum $\Delta S_m(T)$ value of 4.1 J (kg K)$^{-1}$ at $T_N$ and a refrigerant capacity value of 56.6 J kg$^{-1}$.

**Keywords:** Yb magnetism; magnetic phase diagram; magnetocaloric effect; metamagnetism; heavy fermions; Weyl semimetal; first-order magnetic transition; separation of magnetic contributions

## 1. Introduction

It has recently been predicted that the compounds GdCoC$_2$, GdNiC$_2$, NdRhC$_2$ and PrRhC$_2$ are topological Weyl semimetals (TWS) [1]. In these compounds, the inversion symmetry and the time reversal symmetry are violated due to the noncentrosymmetric orthorhombic structure of the CeNiC$_2$-type and the low-temperature magnetic ordering of these compounds, respectively. The unique properties of magnetic TWS can be extremely useful in the context of information technology (e.g., quantum computing), given that such massless charged particles will carry electric current without Joule heating [2]. The theory provides a clear guide to the implementation of magnetic TWS, but, so far, there are only a few experimentally confirmed examples of TWS with time-reversal symmetry breaking [3,4]. It was shown [5] that the YbCoC$_2$ compound has special points with linear dispersion in the electronic band structure (see Figure 1). This allows us to consider this compound as a representative member of the class of TWS with a CeNiC$_2$-type crystal structure.

*R*CoC$_2$ compounds with magnetic rare-earth atoms *R* are ordered ferromagnetically (FM) at low temperatures [6–9]. It is believed that, in these compounds, Co ions do not have a magnetic moment, and the Ruderman–Kittel–Kasuya–Yosida exchange interaction makes a significant contribution to the stabilization of magnetism [10].

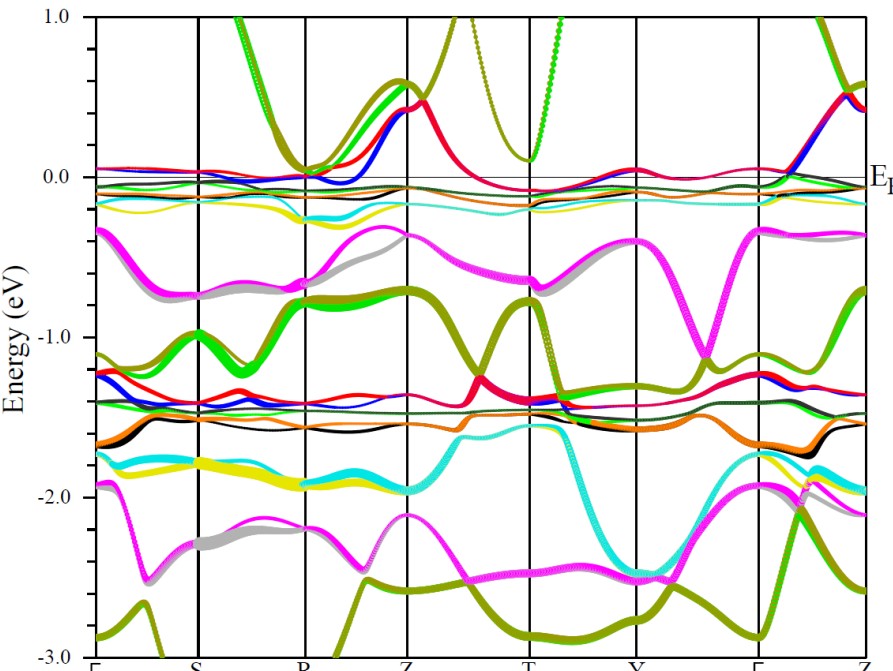

**Figure 1.** The DFT band structure of YbCoC$_2$. The narrow $4f$-Yb states with $j = 7/2$ are present at the Fermi level ($E_F$). The thickness of the lines depicting the bands is proportional to the contribution of the Co states. The colors of the lines are to help visualization .

One of the promising applications of rare-earth-based magnets is their use as cooling refrigerators. This application is possible due to the large magnetocaloric effect (MCE) observed in these magnets. The MCE is the result of a change in the entropy of magnetic spins ($\Delta S_m$) under the action of a magnetic field and can be fully characterized by a change in temperature in an adiabatic process ($\Delta T_m$) or a change in the magnetic entropy in an isothermal process ($\Delta S_m$). A positive MCE leads to sample heating under adiabatic conditions. This property is widely used in low-temperature physics to obtain millikelvin temperatures by demagnetizing a paramagnetic salt under adiabatic conditions. This method is known as adiabatic demagnetization. $\Delta S_m$ can be obtained indirectly from isothermal measurements of the magnetization. Previously, large reversible MCEs in HoCoC$_2$, ErCoC$_2$ [11] and TbCoC$_2$ [12] and giant reversible MCEs in TWS GdCoC$_2$ [13] have been observed using such measurements.

The moderately heavy fermion compound YbCoC$_2$ ($\gamma$ = 190 mJ/mol-K$^2$) has an antiferromagnetic (AFM) transition at $T_N$ = 27 K, which is the highest temperature among the Yb-based magnetic compounds known to date. The magnetic structure of YbCoC$_2$ in a zero magnetic field at $T$ = 1.3–27 K is a sine-modulated incommensurate structure with a wave vector of $(0, 0, k_z)$, where $k_z$ depends on temperature: $k_z = 0.28$ at $T_N$ and locks-in to the value of $k_0 = 1/4$ below $T_{NC}$ = 9 K [5]. This simple change in the wave vector with temperature is characteristic of modulated magnetic structures [14]. At $T = T_{NC}$, there is a rather smooth transition from a commensurate structure to an incommensurable one [5]. The amplitude of the magnetic moment of the Yb ions in a sinusoidal wave is $\mu_{Yb} = 1.32 \ \mu_B$ at 1.3 K, which is much less than the total magnetic moment of the free ion Yb$^{3+}$ ($gJ = 4 \ \mu_B$). This reduced value can be connected with crystal-field splitting of $j = 7/2$ multiplet, Kondo screening of a magnetic moment [5,15] or some covalence in the chemical bonds between Yb and C, which leads to a strong suppression of the effective magnetic moment [16]. We roughly estimate the crystal-field splitting between the $e_g$ and $t_{2g}$ Co bands as occurring at 1.3 eV. The splitting of $4f$ sub-bands can be approximately assessed from the band structure near the $\Gamma$ point (see Figure 1).

In this work, we present the results of studies of the magnetic $H$-$T$ phase diagram and MCE in the YbCoC$_2$ compound in the field range of 0–9 T and temperature range of 2–80 K.

Since YbCoC$_2$ has an AFM ground state, we can expect a richer phase diagram than that for the other $R$CoC$_2$ compounds with simple FM structures. Despite the well-known fact that AFM compounds usually have weak negative MCE, i.e., the material cools when a field is applied, the metamagnetic transition in YbCoC$_2$ leads to a change in the sign of the MCE.

Additionally, as will be shown below, using the Banerjee criterion, we determined that this transition is a transition of the first kind (at least in the magnetic field). Previously, it was shown that a metamagnetic transition in an AFM compound can lead to a large positive MCE [17,18]. The magnetic transition temperature of about 27 K makes it possible to use YbCoC$_2$, for example, to liquefy hydrogen, which is now actively used for various engines [19,20].

## 2. Materials and Methods

A polycrystalline single-phase sample of YbCoC$_2$ was synthesized using a high pressure–high temperature technique at $P = 8$ GPa and $T = 1500$–$1700$ K in a toroid high-pressure cell by melting the Yb, Co and C components and was characterized in Ref. [5]. The Rietveld analysis of X-ray and neutron diffraction patterns shows that the compound crystallizes in an orthorhombic structure of the CeNiC$_2$-type (space group $Amm$2, No. 38), similar to other heavy rare-earth carbides, $R$CoC$_2$. The impurity of the high-pressure phase of non-magnetic ytterbium oxied (YbO) with a fraction of less than 5 wt % was also found in the sample [5].

Magnetic moment measurements were carried out on the VSM option of PPMS (Quantum Design). The isothermal magnetization curves were obtained by increasing the magnetic field from 0 to 9 T and changing the temperature from 2 to 80 K under field cooling conditions with variable temperature steps: $\delta T = 4$ K above and well below $T_N$ and $\delta T = 2$ K near $T_N$. The magnetic field step was held at $\mu_0 \delta H = 0.1$ T.

Ab initio density functional theory (DFT-LDA) calculations of YbCoC$_2$ were performed using the WIEN2k package [21], with spin-orbit coupling taken into account. The calculations were made at the experimental lattice parameters measured in Ref. [5], with subsequent relaxation of atomic coordinates. The Yb $4f$ electrons were considered valence electrons. Comparison of our DFT-calculated band structure (Figure 1) and density of states (DOS) with the DMFT results [5] shows consistency between them in the position of bands and corresponding DOS peaks. Therefore, we believe that our DFT results sufficiently reliably describe the details of electronic structure.

## 3. Results and Discussion

### 3.1. Magnetic Properties

Some $M(H)$ dependencies from the temperature range 2–80 K are shown in Figure 2. Detailed dependency analysis for $T < T_N$ in weak magnetic fields showed that the $M(H)$ curves were almost linear and the magnetization slightly increased with increasing temperature. Additionally, spontaneous magnetization was absent at all temperatures. This behavior is typical for AFM material. At high magnetic fields ($H \geq H_{c1}$), there was a sharp increase in $M(H)$ associated with the metamagnetic transition. The metamagnetic transition has also been discovered from measurements of $M(H)$ in the range of 6–8 T in previous investigations [5]. For this transition, a hysteresis was observed on the curves $M(H)$ [5] (not shown here). In the paramagnetic region (at $T > T_N$), no metamagnetic transition was observed. Therefore, its nature is associated with the AFM structure of the magnetic moments of Yb.

$M(T)$ measured in a magnetic field of 7 T (see Figure 3) demonstrated a plateau-like behavior at $T \lesssim 24$ K (this temperature was determined as the minimum of the function $dM/dT$ of $T$), which corresponds to the transition from the paramagnetic (PM) to the ferromagnetic state. We have plotted the magnetic $H$-$T$ phase diagram of YbCoC$_2$ obtained from the magnetization measurements (see the inset of Figure 3). $H_{c1}$ and $H_{c2}$ were determined as local maxima in $dM/dH$ vs. $T$ (see right panel of Figure 4) and from $M(T)$ dependencies. $H_{c1}$ was associated with the metamagnetic transition to the intermediate

magnetic phase (IM), and $H_{c2}$ was probably associated with the transition of YbCoC$_2$ to the FM state induced by the external magnetic field in which the magnetic unit cell has a finite value of magnetization.

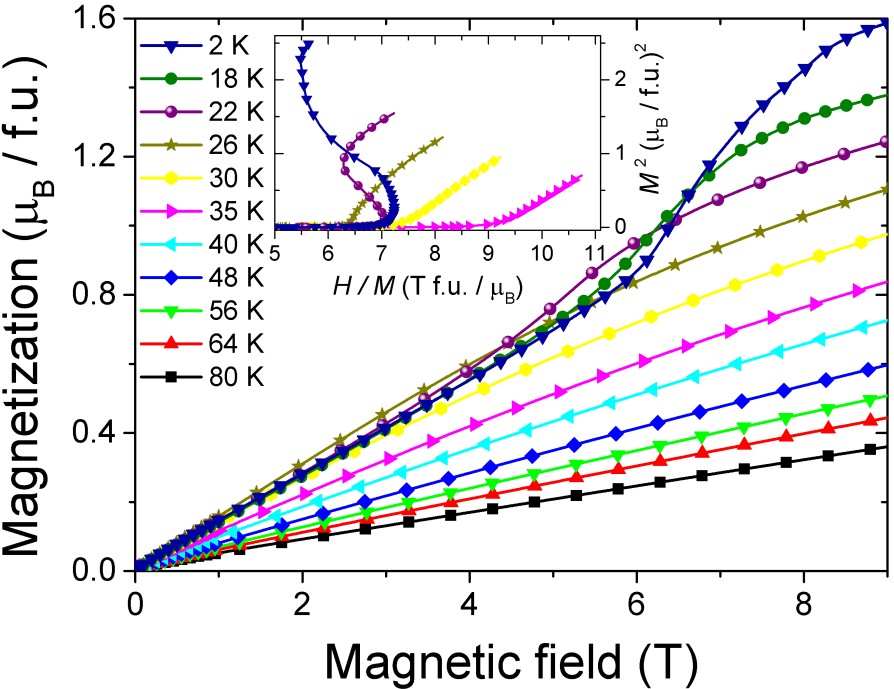

**Figure 2.** Isothermal magnetization dependencies $M(H)$ of YbCoC$_2$. Inset: Arrot plots ($M^2$ vs. $H/M$) for selected temperatures.

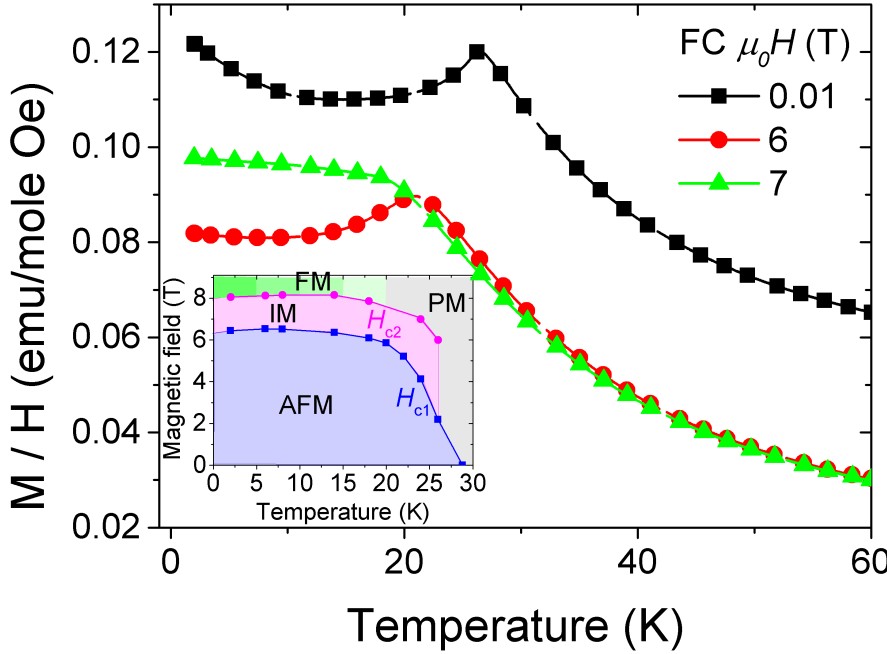

**Figure 3.** Temperature dependencies of $M/H$ measured in various external magnetic fields in field-cooled mode. Inset: Possible magnetic $H$-$T$ phase diagram of YbCoC$_2$ (where AFM—antiferromagnetic phase (blue area), IM—intermediate magnetic phase corresponding to the metamagnetic phase transition (pink area) and FM—ferromagnetic phase (green area)).

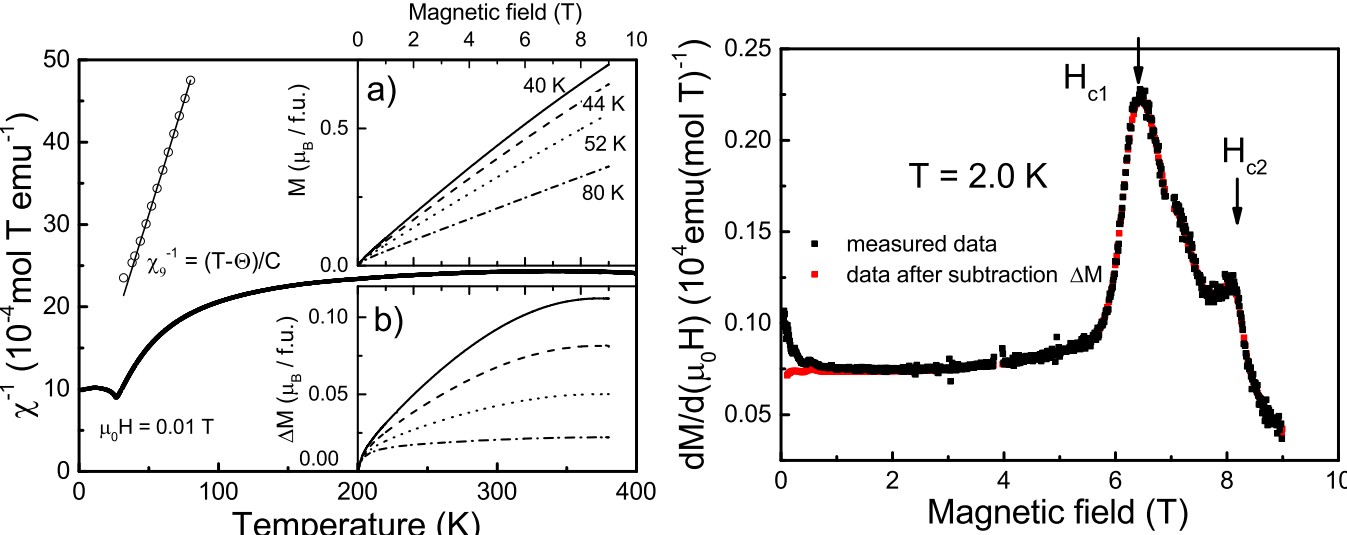

**Figure 4.** Left panel: Black curve is a reciprocal magnetic susceptibility $\chi^{-1}$ of YbCoC$_2$ compound measured at H = 100 Oe; open circles are $\chi_9(T)$ obtained as a slope of $M(\mu_0 H)$ curves at 9 T at different $T$; solid line is a Curie–Weiss fit $\chi^{-1} = (T - \Theta)/C$, with $\Theta = -7.3$ K and $C = 1.84$ emu K/(Oe mol). Insets a) and b): magnetization curves $M(H)$ of YbCoC$_2$ and the difference $\Delta M = M(H) - \chi_9 \cdot H$ at 40, 44, 52 and 80 K, shown in the, respectively. The abscissa axes coincide on insets a) and b). Right panel: The first derivative of the original $M(\mu_0 H, 2)$ (black) and corrected $[M(\mu_0 H, 2) - \Delta M(\mu_0 H, 80)]$ (red) magnetization curves at $T = 2.0$ K. The vertical arrows indicate the peaks that determine the magnetic fields $H_{c1}$ and $H_{c2}$.

Some Arrot curves have a negative slope (see the inset of Figure 2). According to the Banerjee criterion [22], this indicates that the observed metamagnetic transition belongs to the first order.

The saturation of magnetization is not observed in fields of 9 T at all temperatures. $M(9 \text{ T}) \approx 35$ J (T kg)$^{-1}$ at $T = 2$ K, which corresponds to a magnetic moment of about 1.6 $\mu_B$/f.u. This value is smaller than the saturation magnetic moment of the Yb$^{3+}$ ion ($m_s = 4.0\mu_B$).

Since the magnetic structure of the YbCoC$_2$ compound at $T = 2$ K and at zero magnetic field is sinusoidal (see Figure 5) [5], we note that the average value of the magnetic moment of the Yb atom in, for example, the positive period of this sinusoidal structure containing 4 Yb atoms is $1.32 \cdot (1 + 2 \cdot sin(\pi/4)) / 4 = 0.8 \mu_B$ / f.u. It is interesting to note that $M(H)$ for $T = 2$ K reaches this value exactly at $H_{c1}$. The simplest explanation for this behavior of the magnetization can be the following assumption. Here, when the field is rising $H = 0$–$H_{c1}$, there is a smooth transition from sinusoidal modulation to a spin-polarized magnetic structure, which consists of the rotation of all magnetic moments along the field with the conservation of the average magnetic moment on Yb sites with an increasing field without changing their values (see Figure 5). Such a behavior of the magnetic moments, which is possible in the case of overcoming the magnetocrystalline anisotropy by the magnetic field, will give the observed dependence of the magnetization.

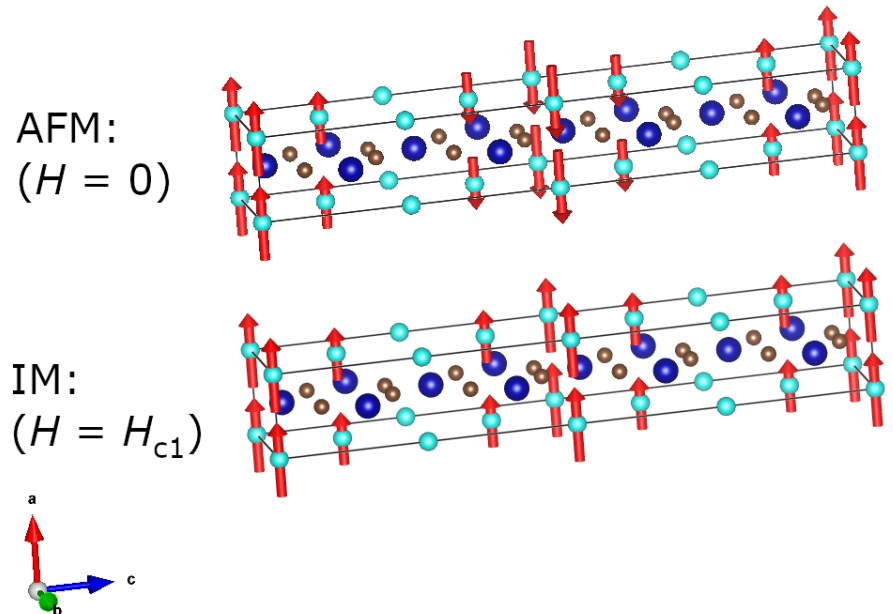

**Figure 5.** Possible transformation of YbCoC$_2$ magnetic structure in the external magnetic field: from the antiferromagnetic (AFM) sine wave at zero field to the intermediate spin-polarized (IM) at $H = H_{c1}$ (generated using the VESTA software [23]). The Yb (cyan circles) and their magnetic moments (red arrows) and Co (blue) and C (brown) ions are shown.

As seen from Figure 2, the $M(H)$ exceeds the value of 0.8 $\mu_B$/f.u for $H > H_{c1}$ in the magnetically ordered state. So, the mean magnetic moment increases in value above $H_{c1}$, which may be connected with the increase in the Yb mean magnetic moment and the additional contribution from Co.

Inset (a) of the left panel of Figure 4 shows the magnetization curves at some selected temperatures. In view of the not-completely paramagnetic behavior of these curves expected at these temperatures, we classified them into paramagnetic $\chi_9(T) \cdot H$ and nonlinear contributions $\Delta M(\mu_0 H, T)$, shown in inset (b) of the left panel of Figure 4. The susceptibility $\chi_9(T)$ was obtained as the slope of the magnetization curve $M(\mu_0 H, T)$ at $\mu_0 H = 9$ T and $\Delta M(\mu_0 H, T) = M(\mu_0 H, T) - \chi_9(T) \cdot \mu_0 H$. The obtained inverse susceptibility $\chi_9^{-1}(T)$ demonstrated a strict linear dependence on temperature in the measured PM range of 38–80 K, and is shown in the left panel of Figure 4 (open circles). The dependence $\chi_9^{-1}(T)$ was fitted by the Curie–Weiss law $\chi^{-1} = (T - \Theta)/C$ (solid line Figure 4 left panel), where the negative Curie temperature $\Theta = -7.3$ K indicates that the main interaction is AFM. The estimated effective magnetic moment was $\mu_{eff} = 3.84$ $\mu_B$, which is less than the effective moment for the free ion Yb$^{3+}$ (4.54 $\mu_B$).

The nonlinear contribution to the magnetization $\Delta M(\mu_0 H, 80)$ at $T = 80$ K was preserved over a fairly wide temperature range. The subtraction of this contribution from the magnetization curves $M(\mu_0 H, T)$ led to their smoothing. The effect of such a procedure was much more pronounced on the derivative $dM(\mu_0 H, T)/d(\mu_0 H)$ than on the curve $M(\mu_0 H, T)$ itself. For example, the right panel of Figure 4 shows the first derivatives of the original $M(\mu_0 H, 2)$ (black points) and the corrected $M(\mu_0 H, 2) - \Delta M(\mu_0 H, 80)$ (red points) curves. As a result, the anomaly in the weak field was eliminated. Additionally, the magnetization changed almost linearly with the field, up to about $H_{c1}$. The linear behavior of magnetization corresponds to the smooth rotation of magnetic moments and their alignment along the field. A similar result was also observed for the remaining $M(\mu_0 H, T)$ curves at other temperatures. As is shown elsewhere [24], this $\Delta M(\mu_0 H, 80)$ magnetic contribution, which is ordered in low fields, persisted as the temperature rose by much more than the 80 K presented here (see Figure 4, left panel), namely up to at least $T = 400$ K. We think that this magnetic contribution is external to YbCoC$_2$ and may

be attributed to the small fraction ($\approx$0.3 %) of magnetic impurities. A slight decrease in $\chi^{-1}(T)$ (see Figure 4, left panel) may also be connected with additional magnetism caused by such impurities.

### 3.2. Magnetocaloric Effect

$\Delta S_m(T)$ and its relative error for different $\Delta H = H_f - H_i$ were determined from the isothermal magnetizations using well-known numerical methods [25].

$$\Delta S_m(T) = \int_{H_i}^{H_f} \left(\frac{\partial M}{\partial T}\right)_H dH, \tag{1}$$

where $H_i$ (= 0 in the present case) and $H_f$ are the initial and final values of the applied magnetic fields, respectively. The relative error of $\Delta S_m(T)$ did not exceed 10%.

The dependencies $-\Delta S_m(T)$ for $\Delta H \leq$ 0–6 T have segments with positive and negative MCEs (see Figure 6). These dependencies have minima, which are connected with the AFM nature of the magnetic ordering in YbCoC$_2$ in low magnetic fields. The $-\Delta S_m$ minima shift towards lower temperatures with an increase in $\Delta H$, in accordance with the phase diagram (the inset of Figure 3), and the positive MCE appears at $T < 10$ K for $\Delta H \geq$ 0–5 T. For $\Delta H = $ 0–9 T, a strong external magnetic field suppresses the AFM structure, MCE becomes positive in the full temperature range, and $-\Delta S_m(T)$ has a typical FM "caret-like" shape [26]. $-\Delta S_m(T)$ reaches a maximum of 4.1 J/kg-K at $T \approx 28$ K.

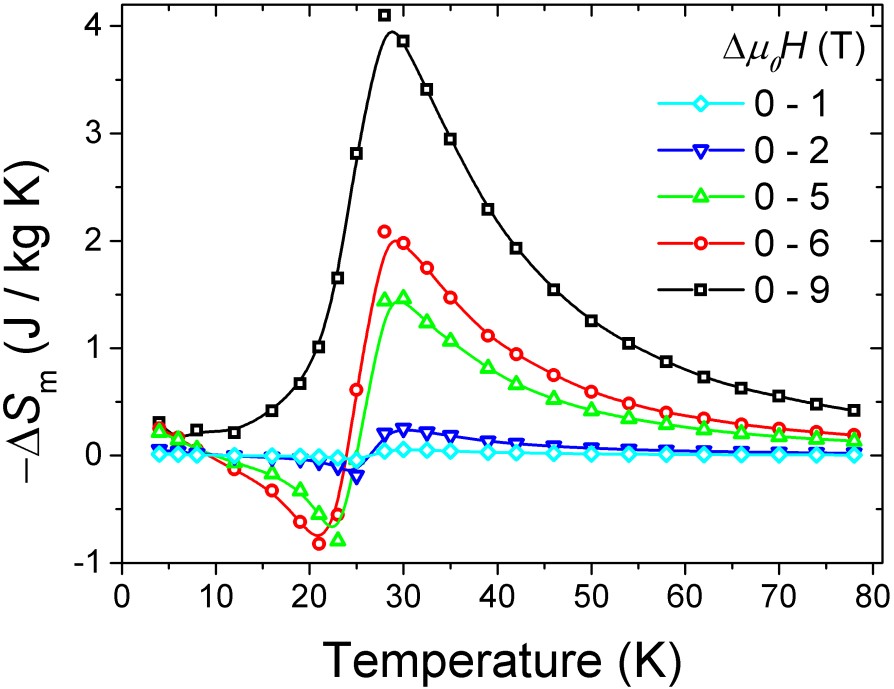

**Figure 6.** Temperature dependencies of the magnetic entropy change $-\Delta S_m(T)$ for YbCoC$_2$ for different magnetic field changes (points — experimental data, lines — spline fits ).

The amount of heat that can be transferred between the cold and hot parts in one cooling cycle is RC = $\int_{T_1}^{T_2} |\Delta S_m| dT$ = 56.6 J/kg for $\Delta H = $ 0–9 T. Here, $T_1$ and $T_2$ are the temperatures corresponding to both sides of the half maximum value of $-\Delta S_m(T)$.

The obtained value of the MCE in YbCoC$_2$ is rather low in comparison with the other $R$CoC$_2$ compounds ($R$ = Gd, Tb, Ho, Er). This may be attributed to the lower value of the Yb magnetic moment and high magnetic fields, which are necessary to obtain the metamagnetic transition in YbCoC$_2$. If we compare the $-\Delta S_m$ maxima for two TWS, YbCoC$_2$ and GdCoC$_2$, we find max($-\Delta S_m$)/$Rln(2J + 1)$ = 0.18 and 0.46, respectively.

The MCE is rarely studied for Yb-based magnets, except in situations where it is required to obtain extremely low temperatures of the order of 0.1 K [27,28].

## 4. Conclusions

In summary, we established the magnetic $H$-$T$ phase diagram of $YbCoC_2$ in the range of magnetic fields 0–9 T and the temperature interval 2–30 K. It is shown that, with an increase in the external magnetic field, the metamagnetic transition of $YbCoC_2$ to the IM phase occurs, and with a further increase in the magnetic field, it goes to the FM phase. We have determined that $YbCoC_2$ exhibits a first-order transition at a metamagnetic transition below a magnetic ordering temperature of $T_N = 27$ K. The magnetic structure of the IM and FM phases requires further research by means of neutron diffraction in the external magnetic field and magnetization measurements on a single crystal. The smooth transformation from the sine wave modulation to the spin-polarized magnetic structure for $H = 0$–$H_{c1}$ was observed. A further increase in the magnetic field led to an increase in the average magnetic moment Yb and an additional contribution from the magnetic moment Co. No magnetic saturation of the magnetic moment was observed at 9 T and 2.0 K. A small magnetic contribution from the possible $Co^{2+}$ on the order of 0.3% was found and isolated. The MCE for $YbCoC_2$ has been calculated for $\Delta H$ up to 9 T. Due to the AFM–FM transition, the MCE in $YbCoC_2$ changed sign with the increase in $\Delta H$.

**Author Contributions:** Conceptualization, D.A.S., A.V.T., A.V.; methodology, D.A.S., V.N.K.; formal analysis, V.N.K., D.A.S.; resources, A.V.B., A.V.T.; data curation, V.N.K., A.V.S.; vizualization, D.A.S., V.N.K.; calculation, M.V.M.; writing—original draft preparation, D.A.S., V.N.K. and A.V.S.; supervision, A.V.T.; funding acquisition, A.V., Z.S. and A.V.T. All authors have read and agreed to the published version of the manuscript.

**Funding:** This experimental research was funded by the Russian Science Foundation Grant No. 22-12-00008. We are grateful for the support in performing magnetization measurements provided by the Polish representative at the Joint Institute for Nuclear Research.

**Data Availability Statement:** The data presented in this study are available on request from the corresponding author.

**Acknowledgments:** The authors are grateful to V. V. Brazhkin for support of the work, P. V. Enkovich for the help in samples characterization, and M. A. Anisimov for fruitful discussions.

**Conflicts of Interest:** The authors declare no conflict of interest. The funders had no role in the design of the study; in the collection, analyses or interpretation of data; in the writing of the manuscript; or in the decision to publish the results.

## Abbreviations

The following abbreviations are used in this manuscript:

| | |
|---|---|
| TWS | topological Weyl semimetals |
| FM | ferromagnetic |
| MCE | magnetocaloric effect |
| AFM | antiferromagnetic |
| VSM | vibrating sample magnetometer |
| PPMS | physical properties measurement system |
| PM | paramagnetic |
| IM | intermediate magnetic phase |
| DFT | density functional theory |
| DOS | density of states |
| DMFT | dynamical mean field theory |

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
