# Peer review of "Some Magnetic Properties and Magnetocaloric Effects in the High-Temperature Antiferromagnet YbCoC2"

_magnetochemistry, doi:10.3390/magnetochemistry9060152_

Round 1

Reviewer 1 Report

The article reports on the H-T magnetic phase diagram and on the magnetocaloric effect in the moderately heavy fermion compound YbCoC2, in the range of magnetic fields 0–9 T and the temperature interval 2–30 K. The investigations results come in addition to their previous study (Salamatin et al., PRB 101 100406 2020).  

The article is well written, containing very good structured information with valuable scientific content. As consequence, the reviewer is recommending its publication in the Magnetochemistry journal.

Some minor corrections: 

P1 row33: please reformulate ‘depending on from a’

P2 row 84: Fig 4 is called before Figs 2 and 3. If possible, re-ordering of the figure numbers would be recommended.  

P3 row 97 “J/T-kg” for the units of magnetization comes with additional line.

P7 row 176 Please reformulate the sentence “While further increases in magnetization in higher magnetic…”

Only minor editing is required. 

Author Response

Dear Editor and Referee,

We would like to thank you for your great effort and your helpful and positive comments concerning our manuscript.

We have corrected our manuscript in accordance with the referee’s comments. In what follows, we give detailed responses to the comments and indicate the changes made to the manuscript. We believe that our present manuscript is now suitable for publication in the Magnetochemistry.

 Yours sincerely,

Dr. Denis Salamatin on behalf of the authors.

1) P1 row33: please reformulate ‘depending on from a’

We removed this text from the manuscript. We agree that this part is redundant (page 2, paragraph 1, line 35):

depending on from a change in an external magnetic field.

2) P2 row 84: Fig 4 is called before Figs 2 and 3. If possible, re-ordering of the figure numbers would be recommended. 

We reordered the figures according to their references in the manuscript.

3) P3 row 97 “J/T-kg” for the units of magnetization comes with additional line.

Now the units of magnetization (μB / f.u) is on one-line.

4) P7 row 176 Please reformulate the sentence “While further increases in magnetization in higher magnetic…”

We reformulated this sentence (page 8, paragraph 1, line 198-199):

"While further increases in magnetization in higher magnetic field is connected 
with the increase. A further increase in the magnetic field leads to an increase ..."

Reviewer 2 Report

This paper describes the studies of the magnetic H-T phase diagram and MCE in the YbCoC2 compound in the field range of 0 – 9 T and temperature range of 2 – 80 K.  An increase in the external magnetic field, the metamagnetic transition of YbCoC2 to the IM phase occurs, and with a further increase of magnetic field, it goes to the FM phase. YbCoC2 exhibits a first order transition at metamagnetic transition below magnetic ordering temperature of TN = 27 K. The manuscript is well prepared and I recommend this paper for publication in the Magnetochemistry in the present form.  

Author Response

Dear Editor and Referee,

We would like to thank you for your great effort and your helpful and positive comments concerning our manuscript.

 Yours sincerely,

Dr. Denis Salamatin on behalf of the authors.

Reviewer 3 Report

This manuscript reports the magnetic and magneto-caloric properties of poly-crystalline bulk YbCoC2. The characterizations presented can aid a deeper understanding of magnetism in rare-earth and transition metal ceramics. The paper is well written and, with minor revisions, can be recommended for publication:

1)   In the first paragraph of the second page, additional reasons for a lower Yb magnetic moment could also be the Fock exchange mechanism and the more covalent bonds between heavier cations and coordinating anions. These points should be mentioned and discussed as per the following papers:

Pham et al. Critical role of Fock exchange in characterizing dopant geometry and magnetic interaction in magnetic semiconductors, PHYSICAL REVIEW B 89, 155110 (2014); https://doi.org/10.1103/PhysRevB.89.155110

Streltsov et al. Covalent bonds against magnetism in transition metal compound, PNAS 113, 10491-10496 (2016); https://doi.org/10.1073/pnas.1606367113

2)   Can the authors present the electronic structure, in terms of d and f orbitals splitting, for both Yb and Co under the crystal field in this material?

3) Figure 2 should be improved to contain all elements, so the coordination environment of the magnetic ions can be quickly visually inspected.

4) Figures 4 and 5 should be placed before the conclusion section.

5) Ideally, a phase characterization such as XRD and a microscopic image such as SEM of the sample should also be included.

The language style is adequate.

Author Response

Dear Editor and Referee,

We would like to thank you for your great effort and your helpful and positive comments concerning our manuscript.

We have corrected our manuscript in accordance with the referee’s comments as much as possible at this time. In what follows, we give detailed responses to the comments and indicate the changes made to the manuscript. We believe that our present manuscript is now suitable for publication in the Magnetochemistry.

 Yours sincerely,

Dr. Denis Salamatin on behalf of the authors.

1)   In the first paragraph of the second page, additional reasons for a lower Yb magnetic moment could also be the Fock exchange mechanism and the more covalent bonds between heavier cations and coordinating anions. These points should be mentioned and discussed as per the following papers:

Indeed, such approaches as GW approximation or hybrid functionals can provide more accurate results compared to standard DFT. However, the more appropriate method to treat strongly correlated rare-earth compounds (like YbCoC2) is the dynamical mean field theory (DMFT), in particular, LDA+DMFT as implemented, e.g., in the eDMFT package. The DMFT method is known, among other things, for very accurate determining the local coordinates of atoms surrounding the correlated atom. Nowadays, the methods combining DMFT and HSE functionals are developed [Arpita P. et al., Annual Review of Materials Research 49, 31 (2019)], however, these approaches are highly computationally demanding. The LDA+DMFT method has been previously employed for YbCoC2 in [5]. Thus, we performed DFT-LDA calculations of YbCoC2 and found a reasonable consistency with the results [5]. Now our calculations are described in the revised manuscript. We approximately estimate d- and f-orbitals splitting on the basis of these results. 

As is known from many investigations of transition-metal and rare-earth carbides,
these compounds exhibit a mixed bonding. YbCoC2 is metallic, and as such, has a metallic bonding with some admixture of covalency in the metal-carbon bonds. To determine the degree of covalency, additional calculations are needed, for example, of the electron localization function (ELF).

It is well known that in the lanthanides the degree of covalent bonding is very weak in comparison with d-block transition metals [Titova et al. Communications Chemistry 5, 12 (2022)]. Usually the reduced magnetic moment in such systems is described to the crystal field effects. For example, in the tetragonal phase of YbCo2Si2 the ground state of Yb is a doublet with a magnetic moment of ~ 1.8 μB per Yb [C. Klinger et al., New Journal of Physics 13, 083024 (2011)].

Thank you for the comment we will look into this possibility of reducing the magnetic moment carefully.

We added results of DFT calculations (Figure 1) and some discussions about the redution of magnetic moment (page 2, paragraph 2, lines 52-54):

"This reduced value can be connected with crystal-field splitting of j = 7 / 2 multiplet, Kondo screening of magnetic moment [5, 15] or some covalence in chemical bonds between Yb and C, which leads to a strong suppression of the effective magnetic moment [16]."

2) Can the authors present the electronic structure, in terms of d and f orbitals splitting, for both Yb and Co under the crystal field in this material?

The accurate calculations of 4f CEF splitting require significant resources and time and should be checked with expertimental techniques. We estimated d- and f-orbitals splitting on the basis of DFT results and add the details in the text (page 2, paragraph 2, lines 54-57):

"We roughly estimate the crystal-field splitting between the eg and t2g Co bands as 1.3 eV. The splitting of 4 f sub-bands can be approximately assessed from the band structure near the Γ point (see Fig. 1)."

3) Figure 2 should be improved to contain all elements, so the coordination environment of the magnetic ions can be quickly visually inspected.

We added all ions (+ Co and C) on Figure 3 (previously Figure 2).

4) Figures 4 and 5 should be placed before the conclusion section.

We placed the Figures before the conclusion section.

5) Ideally, a phase characterization such as XRD and a microscopic image such as SEM of the sample should also be included.

The phase characterization with XRD and neutron powder diffraction of the studied sample was performed in Ref. [5]. We have SEM images of the crystals (see one in the attachment) and we could add it to the manuscript. But in my opinion such figures very typical for the samples and do not carry much useful information, since they depend on the cleavage of the sample. Also it is known that it is difficult to determine accurately the atomic concentration of carbon in the sample from EDS due to the low Z and high absorption of the corresponding X-rays. The atomic concentrations of Yb and Co are approximately equal.

We added in the text the details about phase characterization (page 3, paragraph 2, lines 74-78):

"The Rietveld analysis of X-ray and neutron diffraction patterns shows that the compound crystallizes in an orthorhombic structure of the CeNiC2-type (space group Amm2, No. 38), similar to other heavy rare-earth carbides RCoC2. The impurity of high-pressure phase of non-magnetic ytterbium oxied (YbO) with a fraction of less than 5 wt. % was also found in the sample [5]."
